# Partnering with the Health Professions to Promote Prevention of an Alcohol-Exposed Pregnancy: Lessons Learned from an Academic–Organizational Collaborative

**DOI:** 10.3390/ijerph16101702

**Published:** 2019-05-15

**Authors:** Leigh Tenkku Lepper, Diane King, Joy Doll, Sandra Gonzalez, Ann Mitchell, Joyce Hartje

**Affiliations:** 1School of Social Work, University of Missouri, Columbia, MO 65211, USA; 2Center for Behavioral health Research and Services, University of Alaska Anchorage, Anchorage, AK 99508, USA; dkking@alaska.edu; 3Center for Interprofessional Practice, Education and Research, Creighton University, Omaha, NE 68178, USA; joydoll@creighton.edu; 4Department of Family and Community Medicine, Baylor College of Medicine, Houston, TX 77098, USA; Sandra.Gonzalez@bcm.edu; 5School of Nursing, University of Pittsburgh, Pittsburgh, PA 15261, USA; ammi@pitt.edu; 6Center for the Application of Substance Abuse Technologies (CASAT), University of Nevada, Reno, NV 89557, USA; jhartje@casat.org

**Keywords:** alcohol-exposed pregnancy, screening and brief intervention, social–ecological model, academic–professional organization collaborative

## Abstract

*Background*: Evidence-based strategies exist to train healthcare professionals to ask their patients and clients about alcohol use, and are successful. Implementation of these strategies utilizing a system-level approach has not been conducted nationwide. This case study reports on the success of academic partnerships with national health professional organizations to increase adoption of evidence-based strategies to prevent alcohol-exposed pregnancies. *Methods*: Authors reviewed and summarized multi-level strategies created as part of the developmental phase of this project in order to report successes and challenges. We applied the three principles of reflection, sense-making, and reciprocal learning, as identified in the practice change literature, to synthesize our experience. *Results*: There were five primary lessons learned as a result of this work: Development of technology-based training websites requires significant time to design, implement, and test; project ‘mission-drift’ is inevitable, but not necessarily unwelcome; time and effort is required to create and sustain functioning workgroups when there are different organizational cultures; and changing real-world practice is hard to do, yet changing the conversation on screening and brief intervention is possible. *Conclusions*: Use of multi-level strategies within an academic–professional organization model was successful in promoting awareness and education of healthcare professionals in the prevention of alcohol-exposed pregnancies.

## 1. Introduction

Although the exact prevalence of Fetal Alcohol Spectrum Disorders (FASDs) is unknown, it is believed that up to one in every five U.S. school children may fall into the spectrum [1]. In addition, one in nine women continue to report drinking alcohol during pregnancy [2]. Despite recognition that consuming alcohol during pregnancy has the greater effect of harm to a developing fetus compared to other drugs [3], healthcare professionals still vary on discussing alcohol abstinence during pregnancy for a variety of reasons, including lack of knowledge or skepticism regarding current evidence that alcohol use in pregnancy negatively impacts fetal development, concerns that discussing alcohol use will increase their patients’ anxiety or guilt, concerns about legal protection of their patients’ privacy [4], and their own assumptions about which groups of patients are at risk [5,6,7]. In one study, as many as 17% of the physicians surveyed believed “occasionally” drinking while pregnant was acceptable [8]. The literature clearly indicates a need to provide healthcare professionals with clear and consistent information on the risks associated with any alcohol use during pregnancy, and on the evidence-based strategies that can be used in their practice to ensure that all women have the information and support they need to promote an alcohol-free pregnancy and prevent FASDs. Multiple healthcare professionals may encounter a woman while she is pregnant, offering numerous opportunities to engage in or reinforce discussions about alcohol use and its risks during pregnancy. Reaching students and currently practicing healthcare professionals with up-to-date information, resources, and training is an important strategy for addressing the public health concern of prevention of FASDs. This article will describe the development, implementation, and results of a national collaborative model designed to provide healthcare professionals with training and materials for use in speaking to their clients and patients on prevention of an alcohol-exposed pregnancy. All activities related to the project were approved by the institutional review board for each of the participating universities.

### 1.1. From Training to Practice Change: Evolution of the Academic–Organizational Model

In our previous work as Centers for Disease Control and Prevention (CDC)-funded regional training centers (RTCs), the focus was on training healthcare professionals to prevent, identify, diagnose, and treat FASDs within diverse geographic regions across the U.S. The RTCs were housed within five academic institutions, and included faculty-led teaching and knowledge courses promoted directly to healthcare professionals. A comprehensive FASDs curriculum was designed to standardize knowledge and skill training, covering seven core competencies established by the CDC [9]. Various strategies were designed by the RTCs to disseminate the training curriculum through their respective regions. 

In a peer-review study on the 11-year impact of the RTCs, reviewers recommended moving from a regional to a national approach, building upon and expanding the regional accomplishments of the RTCs, while achieving national reach. Recommended strategies for national dissemination and implementation included using technology to increase healthcare professional access to training, tools, and resources; and collaboration with medical societies and national healthcare professional associations to encourage their endorsement of best-practice recommendations that would increase knowledge and promote adoption of evidence-based interventions to prevent FASDs. One example of an evidence-based practice change that has been targeted for widespread dissemination is universal alcohol screening and brief intervention (SBI), which has been shown through numerous randomized clinical trials to be effective in helping patients reduce their alcohol consumption [10,11,12]. Based upon decades of evidence for alcohol SBI, the U.S. Preventive Services Task Force continues to recommend that primary care providers ask all their adult patients, including pregnant women, about their alcohol use using standardized screening tools, and provide a brief behavioral intervention to patients engaged in risky alcohol use [13]. However, adoption of alcohol SBI has been slow [14,15]. Another important strategy was to use the national platform and collaborations to disseminate a clear and consistent message that health care providers could relay to their patients, i.e., that there is “no known safe amount, no safe time, and no safe type of alcohol” to consume while pregnant [16].

### 1.2. Overview of the Project

#### 1.2.1. Formation and Goals of the Academic–Organizational Partnerships

The project being reported on here was supported by the CDC’s National Center on Birth Defects and Developmental Disabilities (NCBDDD) over a four-year period, 2014–2018, with three primary goals: (1) Increase the number of healthcare professionals who receive training and demonstrate improvements in knowledge, attitudes, and skills specific to FASD prevention, early identification, diagnosis, and treatment; (2) collaborate with medical societies and other professional groups in order to expand the reach of FASD-prevention content and resources; and (3) increase the number of healthcare systems that adopt evidence-based interventions/clinical preventive services focused on preventing alcohol-exposed pregnancies.

To achieve these goals, the CDC paired six academic teams, five of which were former regional training center grantees, with a national partner organization (e.g., a medical society or professional association), to form “discipline-specific workgroups”, or “DSWs”. The intent was to focus efforts within the health professional disciplines that were most likely to reach women of reproductive age for primary prevention of alcohol-exposed pregnancies. Five DSWs were focused on prevention of having an alcohol-exposed pregnancy: Family Medicine, Medical Assisting, Nursing, Obstetrics–Gynecology, and Social Work. The sixth DSW, Pediatrics, was focused on teaching pediatricians how to diagnose a suspected Fetal Alcohol Spectrum Disorder. For the purpose of this paper, we only include findings from the five DSWs focused on all three primary goals.

In addition to the DSW teams, three cross-collaboration teams were formed to ensure that all members of the DSWs were represented in the development and dissemination of project materials and resources to achieve the shared goals of the project. The three cross-collaborative teams included the *Online Training and Website Development (“Web Dev”) Workgroup*, the *Evaluation (“Eval”) Workgroup*, and the *Interprofessional (“IP”) Workgroup*. The purpose of the *Web Dev* workgroup was to create course content that would be offered through a consolidated national website and learning management system. Members of the *Web Dev* workgroup included representation from each of the DSW teams as well as CDC members. The purpose of the *Eval* workgroup was to develop an evaluation plan that would both design and implement a set of evaluation measures to assess change over time as a result of taking the online courses, and would also evaluate the impact of the academic–national partner collaborative model. Members of the *Eval* workgroup included representation from each of the DSW teams, CDC members, and an outside contractor for evaluation. The purpose of the *IP* workgroup was to develop online training modules on interprofessional care for the prevention of FASDs, given the importance of interprofessional collaborative care to healthcare delivery [17,18,19]. An interprofessional approach to training is particularly relevant for practice-based interventions such as alcohol SBI, which typically involves a variety of clinical staff in screening, educating, intervening, and possible referral for treatment [20,21]. Members of the *IP* workgroup included representatives from each DSW.

#### 1.2.2. Moving from a Regional to National Focus

The RTCs previously had each created regionally-specific in-person training courses as well as online training modules that were disseminated within their geographic regions. In this project, our charge was to utilize our collective expertise as RTCs in conjunction with the national partners, to design, develop, and implement a consolidated set of training modules that would be housed within a learning management system (LMS) at the CDC training/education website for FASDs (www.cdc.gov/fasdtraining). In addition, DSWs were encouraged to develop multi-level strategies with resources and tools that healthcare professionals could access and use in their practice.

#### 1.2.3. Moving from a Multi-Disciplinary Focus to a Discipline-Specific Focus

An early activity of each DSW was to conduct an environmental scan of existing research, trainings, and practice perspectives for their particular discipline in order to inform their approach to achieving goal one. The scans included compiling and evaluating the relevance and quality of existing resources; identifying gaps and needs; and developing discipline-specific recommendations, multi-level strategies, and action plans [22].

#### 1.2.4. Moving from an Individual-Level Training Focus to a Multi-Level, Social–Ecological Focus

The CDC encouraged the DSWs to intervene at multiple levels, including investigating the use of policy, media, and health system change strategies to influence practice behaviors. Hence, the overall approach was informed by the social–ecological model (SEM) developed by McLeroy et al. [23]. The SEM acknowledges that human behavior and health take place within an environmental context that is comprised of multi-level, inter-related domains [24]. By considering these various levels, the SEM provides a structure for identifying pathways for reaching individuals, groups, and communities for prevention and intervention, using multi-faceted, transdisciplinary approaches [25,26]. In addition, given that people and institutions are located within communities and that all are influenced by policy, targeting all levels of the model simultaneously allowed the DSW efforts to sustain improvements in health outcomes [27].

## 2. Materials and Methods

### 2.1. Document Review to Summarize What We Did

Each author of this manuscript reviewed their DSW environmental scans, action plans, and progress and final reports to list the multi-level strategies produced by each DSW and to summarize reported successes and challenges.

### 2.2. Reflection, Sense-Making, and Reciprocal Learning

To help us synthesize the experience of participating in an academic–organizational collaborative and agree on the extent that the novel DSW structure worked to accomplish the three primary goals, we applied three principles that have been identified in the practice change literature as moderators of practice improvement: Reflection, sense-making, and learning [28]. Reflection can be an individual or group process, and can occur while performing an activity or after its conclusion. In either case, reflection involves stepping back to review what took place, in order to gain perspective. Sense-making involves developing plausible interpretations of ambiguous information or cues. Reciprocal learning is developed iteratively and may involve back-and-forth discussion to agree on discoveries. Applying these three principles, each author individually reviewed and reflected on the written summaries. On a group conference call, the authors discussed and made sense of their individual and collective experiences working within the academic–organizational collaborative structure, and agreed on key lessons learned.

## 3. Results

### 3.1. What We Did

**Strategy development:** Each DSW team designed a set of multi-level strategies that included: Online courses, use of media, published articles within medical societies and professional organizations, hand-outs and flyers for promotional ads, curriculum development, and advocacy programs using discipline-specific champions. In addition, all discipline-specific teams collaborated with the National Organization on Fetal Alcohol Syndrome (NOFAS), whose mission is focused on prevention of alcohol-exposed pregnancies and FASDs as an important public health issue. Each team met with NOFAS staff to develop science-based messaging for each discipline, and to explore the use of social media channels (e.g., Facebook and Twitter) to reach the targeted healthcare disciplines. It should be noted that the majority of trainings maintained a discipline-specific focus, in part due to a desire to meet the continuing education needs of the national organizations’ members. A few DSWs (Medical Assisting, Family Medicine, Nursing) collaborated together to develop interprofessional team-based trainings (e.g., alcohol SBI). In general, due to competing demands plus logistical challenges, collaboration across DSWs for the purpose of developing interprofessional resources was limited.

Table 1 provides comparison of the various DSW strategies developed by the five disciplines (Family Medicine, Medical Assisting, Nursing, Obstetrics–Gynecology, and Social Work) organized by SEM domains of Individual, Interpersonal, Community, and Policy levels. More detailed descriptions of the multi-level strategies follow Table 1.

**Family Medicine:** Baylor College of Medicine partnered with the *American Academy of Family Physicians (AAFP)* to train Family Medicine physicians and residents on alcohol SBI in clinical family practice environments. Strategies developed by this academic–medical society partnership included: The *Alcohol Screening and Brief Intervention in the Primary Care Setting* training curriculum, face-to-face training conducted within select family medicine residency programs, delivery of face-to-face and virtual grand rounds for community health providers, dissemination of flyers and email announcements for the new CDC FASD Training Website, implementation of the “Alcohol Misuse” webpage [29], development and implementation of the AAFP-published manual *Addressing Alcohol Use Practice Manual: An Alcohol Screening and Brief Intervention Program*, creation and dissemination of trainings to guide practitioners on implementation of alcohol SBI programming, creation of “smart phrases” within electronic health record software to increase efficiency of patient encounters regarding alcohol use, development of a champions program for Family Medicine providers, and translation of screening tools and educational material to Spanish language at the request of champions from community health centers, in addition to cross-DSW collaboration with the Mountain Plains Practice and Implementation Center (PIC) to provide discipline-specific training to medical assistants at Baylor Family Medicine clinics.

**Medical Assisting:** The University of Nevada, Reno partnered with the *American Association of Medical Assistants (AAMA)* to form the Mountain Plains Practice and Implementation Center (PIC). The purpose of this collaboration was to train medical assistants (MAs), specifically those MAs who are AAMA-certified. Strategies developed within the academic–professional organization collaboration included: Developing three training curricula (*Introduction to Fetal Alcohol Spectrum Disorders: The Medical Assistant’s Role*, *Preventing Alcohol-Exposed Pregnancies: The Key Role of Medical Assistants*, and *Communication Skills for Medical Assistants: Strategies for FASD Prevention)*; and delivering these trainings and other relevant topics through face-to-face training, online self-paced courses, and webinars. Training of Trainers (ToT) events were held to establish a network of MAs who would be able to train other MAs in alcohol-exposed pregnancies (AEP) and FASD prevention. A *Learning Collaborative Champions Network* was established to support MAs who completed the ToT and provide ongoing opportunities for ongoing learning and networking. The Mountain Plains PIC website was created to house training resources and a toolkit for Medical Assistants, titled, *Enhancing the Role of Medical Assistants: A Toolkit to Increase the Capacity of Primary Care Practice to Reduce Risky Alcohol Use and Prevent Alcohol-Exposed Pregnancies* (under development). In addition, a two-sided screening tool (English and Spanish versions) for use in primary care clinics was co-created with the Baylor Family Medicine Practice and Implementation Center (PIC).

**Nursing:** The University of Alaska, Anchorage, The University of California, San Diego, and the University of Pittsburgh partnered with the *National Association of Nurse Practitioners in Women’s Health (NPWH)* and the American College of Nurse-Midwives (ACNM), and collaborated with the *Association of Women’s Health, Obstetric, and Neonatal Nurses (AWHONN)* and the *International Nurses Society on Addictions (IntNSA)* to train nurses in a variety of clinical settings. Strategies developed included: Development of a champions program for nurses, articles published in professional journals, direct emails to members of national nursing organizations, and community education talks to local groups of nurses. Training activities included incorporating FASD and alcohol SBI content into continuing education programs for nurses, infusion of content into nursing student curricula, presentations at professional, international, national, and regional nursing conferences, and development of numerous webinars, including a seven-webinar curriculum series titled *Optimizing Preconception Health: Preventing Unintended Teratogen Exposure in Reproductive-Aged Women*, available for continuing education credit through the NPWH website. In addition, the nursing DSW created a nurse champions toolkit and resources for nurses, to facilitate their providing information to patients about alcohol use and its relationship to healthy pregnancies. National nursing organization position papers were written, endorsing the message about alcohol abstinence during pregnancy and the role that the nurses and nurse-midwives play in preventing alcohol-exposed pregnancies and FASDs [30]. 

**Obstetrics–Gynecology:** The University of Missouri and the *American College of Obstetricians and Gynecologists (ACOG)* partnered together to train obstetricians–gynecologists (ob–gyns) and residents on how to encourage the ob–gyn provider to talk with their pregnant and non-pregnant patients about alcohol use in the prevention and identification of alcohol-exposed pregnancies [31,32,33,34,35,36]. Courses were developed to increase FASDs-related knowledge, and to improve skills and confidence among ob–gyn providers when discussing alcohol use with their patients. The two online courses created by this partnership included *The Role of the OBGYN and FASD*, written by Dr. Beth Barlet, an ob–gyn provider, and *Alcohol SBI for the Healthcare Professional* that included opportunities for role-playing relevant patient scenarios. An extensive champions program [37] was implemented to train ob–gyns as advocates, trainers, and speakers for FASDs prevention along with the design of a two-sided screening tool (English and Spanish versions) for use in ob-gyn clinics. Additional strategies included electronic invitations sent to resident programs to take the online training, Maintenance of Certification (MOC) approval through the ACOG accreditation board (ABOG) in order to offer continuing education certification, and creation of an e-learning website to house the MOC courses (*www.catalystlearningcenter.com*) and monitor their utilization.

**Social Work**: The University of Texas at Austin, Baylor College of Medicine, and the University of Missouri partnered with the *National Association of Social Workers (NASW)* to address the needs of social workers in practice and students in academic programs in the prevention of alcohol-exposed pregnancies. Strategies developed through this partnership included: Training curricula focused on skills-based learning, specifically, alcohol SBI but also an overview of the CHOICES intervention [38] and ways to incorporate alcohol SBI into clinical practice, particularly in integrated settings; development of a social work champions program; training materials disseminated via webinars, practice-based training, and conference presentations at local, state, national, and international conferences; translation of training materials into Spanish along with the training of field workers in Puerto Rico; the training of social work graduate students at Catalyst Learning Center (http://www.catalystlearningcenter.com) who completed the online *Alcohol SBI Training for the Healthcare Professional* course; the publication of an article in the NASW journal, *Children & Schools*, along with practice-oriented articles in specialty section newsletters for school social work, social workers in the courts, healthcare social workers, and social workers in alcohol, tobacco, and other drug settings; development of key communication strategies for messaging and dissemination of information and materials directed towards the NASW membership; development of webinars providing Continuing Education (CE) credit by connecting with leaders in the specialty practice sections and the design of a webinar for social work field instructors through collaboration with the Council on Social Work Education; development of policy statements in collaboration with NASW staff members; and the inclusion of appropriate language and content within the Adolescent Pregnancy and Parenting and the Adolescent Health policy statements adopted by the NASW Delegate Assembly.

### 3.2. What We Learned

In reflecting on the three primary goals of the study, the authors agreed on several key lessons based on the CDC strategy:


**(Grant Project Goal 1): Increase the number of healthcare professionals who receive training.**


Lesson 1. Technology-based training websites take a very long time to design, test, and implement.

Design of the online educational training modules required collaboration among multiple DSWs, the CDC, and an external web and online educational module developer. Communication across all of these teams towards developing a learning platform that was engaging, easily accessed by healthcare professionals, and sustainable was extremely challenging. In addition to offering courses and resources, the website needed to interface with the CDC Continuing Education (CE) site, where users could complete a course evaluation and post-test, and apply for CE credits. The learning management system (LMS), i.e., the software application used for administration, registration, delivery, tracking, and utilization reporting, required time for pilot testing and modification. Due to these complexities, the training website, originally projected for year two of the project, was not available for use until year four.

Lesson 2. Project ‘mission drift’ is inevitable, but not necessarily unwelcome, as long as you drift in the desired direction.

As it became clear that the training website planned for the project was going to take longer than expected, the DSWs looked for ways to reach health professionals for training. A chief concern was to continue progress towards national reach of the targeted healthcare professionals, with an eye towards disseminating information and resources for preventing AEPs, as opposed to offering in-person training. Revised strategies included identifying meetings and conferences where educational content and resources could be presented or distributed, and developing webinars and other e-training courses that could be promoted and hosted by the national organization partners, archived, and eventually offered through the national consolidated website. While opportunistic trainings were delivered across disciplines, these were used as opportunities to obtain feedback from participants that could inform content for the future website.


**(Grant Project Goal 2): Collaborate with medical societies and other professional groups.**


Lesson 3. Creating functioning workgroups from organizations with different cultures, priorities, and communication styles is like an arranged marriage—requiring patience, hard work, and a little luck.

Academic–Organizational partnerships have long been used to promote stakeholder involvement in strategy development and sharing of evidence-based practices [39]. Determinants of successful partnerships, identified in the literature, include having a clear goal and shared purpose, realistic expectations, defined roles, trust, communication, and continual learning [40]. While all partners shared an interest in preventing alcohol-exposed pregnancy, agreement on plans of action and respective roles between the academic and national partners (professional organizations or medical societies) was challenging. Many of the DSW members had not worked together previously, and thus clarifying roles and understanding the respective priorities and needs of different team members took time, and was further inhibited by the geographic distance between partnering organizations. While the DSWs met regularly via conference call, limited opportunities for face-to-face contact reduced the exchange of information, particularly limiting observation of non-verbal cues that often signal acceptance of, or resistance to, ideas. In a few cases, DSWs required an in-person retreat along with a third-party coach to assist with role delineation and help the academic–national organization partners agree on norms for communicating and reaching consensus. The success of these interventions varied, with early intervention being predictably more effective in resolving issues.


**(Grant Project Goal 3): Increase the number of healthcare systems that adopt evidence-based interventions to prevent alcohol-exposed pregnancies.**


Lesson 4. Changing real-world practice is hard.

Dissemination and implementation of evidence-based interventions within “real-world” healthcare settings is a public health challenge that requires overcoming numerous barriers, including competing priorities, time and skill constraints, reimbursement constraints, and acceptance that the change is necessary and feasible [41,42]. Recruiting healthcare systems that were interested and ready to engage in implementation of alcohol SBI or other clinical preventive services was not something most of the DSWs were prepared to do. Accomplishing systems-level implementation is an interprofessional endeavor that was not well aligned with the discipline-focused structure of the project. While the *IP* workgroup did attempt to create some appropriate trainings that could meet the needs of a health system interested in integrating alcohol SBI into routine practice, most of the DSWs remained focused on goals 1 and 2, reaching healthcare professionals through medical societies and national associations to address knowledge, attitudes, and skills for preventing FASDs.

Lesson 5. While changing practice is hard, changing the conversation is possible.

Several of the DSWs worked together to develop organizational position statements intended to take a clear stand on preventing alcohol-exposed pregnancies and FASDs through abstinence from alcohol during pregnancy. These position statements were widely disseminated through national organization websites, conferences, and organization-specific journals. Most of these position statements also advocated for adoption of evidence-based practices. While policy statements alone were likely insufficient to achieve practice change, they directly reached and provided a “call to action” to the healthcare professionals who were members and affiliates of these organizations.

## 4. Discussion: Did This Model Work?

The development of the national academic–partner collaborative was a novel step forward in the evolution of addressing the need to educate and train healthcare professionals on the prevention of alcohol-exposed pregnancies. As members of the regional training centers, initiated in 2002 to provide education and training to healthcare professionals within U.S. geographic regions, we had made huge strides in developing a comprehensive curriculum focused on FASDs and created a wealth of training courses, educational flyers, and brochures to disseminate our messages. However, the RTC focus on training alone was not sufficient to achieve practice change.

The move from regional training to a national academic–partner collaborative model made sense as a way to consolidate training and resources developed by the RTCs and disseminate them to targeted healthcare professionals through the web-based LMS that was promoted and endorsed by national medical societies and associations. Reflecting on the work of the DSWs, the authors concurred that their collective effort succeeded in promoting awareness and support from national organizations, and reaching their members through the use of multi-level strategies and interventions following the SEM framework.

On the individual level, our strategies were focused on enhancing knowledge, skills, and acceptance of the role that the individual healthcare professional can play in preventing alcohol-exposed pregnancy and FASDs. To that end, we developed discipline-specific, online training modules for family medicine providers, medical assistants, nurses, obstetrics–gynecology providers, and social workers working in a variety of healthcare settings. We also encouraged healthcare professionals within the targeted disciplines to become FASD champions, and distributed articles and resources to enhance their knowledge and confidence.

On the interpersonal level, we were able to incorporate FASD prevention content into continuing education programs for the healthcare professional; presentations at local, regional, state, national, and international professional and scientific conferences; webinars; webinar series; and undergraduate and graduate infusion into resident and student curricula in order to prepare the present and future healthcare workforce. However, developing interprofessional trainings and resources was more limited. One example of success in that area was accomplished by the Mountain Plains PIC, who posited that one way to enhance the role of MAs was to facilitate an interdisciplinary team approach to implementing alcohol SBI. To accomplish this, the Mountain Plains PIC worked collaboratively with the Baylor PIC to train MAs (Mountain Plains focus) and physicians (Baylor focus), nurses, and office staff in primary care clinics in the Houston area.

On the institutional level, recruiting healthcare systems and engaging them in systems-level practice change was not feasible. However, a few DSWs did provide training and technical assistance to healthcare providers interested in adopting alcohol SBI. Nurses were encouraged to talk to their patients about alcohol use, and to advise them to abstain from any alcohol during pregnancy via a practice champion’s tool kit that was distributed at professional conferences and published in one of the professional nursing journals. Family medicine practitioners received training to become champions at Baylor College of Medicine, where attendees were instrumental in promoting continuity of the training at the provider level and garnering support from their own center leadership. Setting up practice-based trainings was a challenge due to time constraints, but they were able to train several health centers within the systems and also provided technical support, including translating patient education materials into Spanish.

On the community level, education was provided to local community groups along with utilization of social media through Facebook and Twitter as mechanisms for reaching local-level constituents. Finally, at the public policy level, a number of the academic–partner collaboratives were able to write position or policy statements specific to their academic discipline via the national organization or medical society affiliation.

### Effectiveness of Discipline-Specific Focus

The idea of creating an academic–national partner collaborative composed of discipline-specific workgroups expanded the capacity of the former regional training centers to more widely disseminate training and resources through the use of discipline-specific online training modules. The DSW approach also encouraged development of champions’ networks and engaged the leaders within the partnering medical societies and national associations in advocating for prevention of alcohol-exposed pregnancies by encouraging patients to abstain from alcohol consumption throughout their pregnancy. Several of the involved organizations published clear position statements on preventing alcohol-exposed pregnancies, and the role that health professionals within their discipline should play in preventing FASDs. The discipline-specific focus, however, was less successful in promoting interprofessional education and collaboration across disciplines. In addition, outside of an additional supplemental project also funded through the CDC, none of the DSWs succeeded in implementing alcohol SBI within healthcare systems during the project period, although practice-based training and technical assistance was offered.

## 5. Conclusions

Based upon the experience of this academic–national partner model, we have several recommendations for future projects:

1. Invest in activities to build strong, working partnerships, with early discussion on three primary questions: How will the partnership perform the day-to-day management of the project, how will the partnership address communication barriers, and how will the partnership deal with conflict? In hindsight, an initial discussion of successful approaches for holding productive conference calls, upfront clarification of roles and expectations for the respective academic and national organization partners, and assistance with identifying effective mechanisms for inter-organizational exchange of information may have enhanced DSW performance.

2. Encourage frequent and timely group reflection on the feasibility of achieving project goals, what is working, and what is not, and revise plans accordingly.

3. If technology-delivered training or interventions are planned, build in enough time to cover inevitable delays related to content development, programming, usability testing, piloting, and modification, particularly for projects involving diverse partners and users, all with varying needs and preferences.

4. Balance discipline-specific education with opportunities to build interprofessional skills, in order to assure that training reflects “real-world” models of interprofessional collaborative practice.

## Figures and Tables

**Table 1 ijerph-16-01702-t001:** Alcohol Exposed Pregnancy Prevention Strategies by Social Ecological Model Domains and Discipline.

	Family Medicine	Medical Assistant	Nursing	OBGYN	Social Work
Individual Level					
Face-to-Face Training	X	X	X		X
Online Training Modules	X	X	X	X	X
Champions Program	X	X	X	X	X
Webinars		X	X		X
Resources/toolkits developed	X	X	X	X	
CE Certification Offered	X	X	X	X	X
Spanish Translation of materials		X			X
Interpersonal Level					
Articles published in discipline-specific journals	X	X	X	X	X
Community Level					
Direct emails			X		
Organizational newsletters		X	X	X	
Content within Discipline-specific Coursework		X	X		X
Content within CE programs		X	X	X	
Utilization of social media		X	X	X	
Attending District and National Conferences		X	X	X	
Policy Level					
Organization position papers	X		X	X	X
Policy Papers within Organization			X		X

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
