# Peer review of "Partnering with the Health Professions to Promote Prevention of an Alcohol-Exposed Pregnancy: Lessons Learned from an Academic–Organizational Collaborative"

_ijerph, 2019, doi:10.3390/ijerph16101702_

Round 1
Reviewer 1 Report
The manuscript entitled "Partnering with the Health Professions to Promote 3 Prevention of an Alcohol-Exposed Pregnancy: 4 Lessons Learned from an Academic-Organizational 5 Collaborative" is a case study report on the methods and results of a CDC funded initiative to increase awareness and discussion of the dangers of drinking while pregnant with females of reproductive age. The report details the means of communication and edusation used and the hurdles faced by the working groups. Overall, a major lesson is the amount of time required to design and implement web based education techniques was very time consuming. Also, creating dialog and a plan of action across various institutes was complex and challenging. The paper is clearly written and appropriately submitted as a case study.
Author Response
Reviewer 1
Thank you for the opportunity to revise and resubmit our manuscript titled, “Partnering with the Health Professions to Promote Prevention of an Alcohol-Exposed Pregnancy: Lessons Learned from an Academic-Organizational Collaborative”, which we hope is now suitable for publication in International Journal of Environmental Research and Public Health as a case report. We deeply appreciate the reviewers’ suggestions and have addressed them as follows:
(1) The content is too long for a case report, simplification should be performed. We have condensed the descriptions of the discipline-specific workgroup strategies and have deleted redundant information in the discussion section.
(2) Please provide the full name of “CDC” for its first appearance. Thank you for catching this error. We now provide the full name for CDC where it first appears (section 1.1).
(3) Was ethical approval needed for this study? While this manuscript does not report any data collected from human subjects, ethical approvals for the overall project were obtained through the Institutional Review Boards associated with each author’s academic institution. We now state this at the end of the Introduction (section 1).
(4) Recommendations 2 to 4 could have stronger discussions. For example, for point 4, what does “real-world” mean? Don’t we already know that public health interventions are related to real world? We agree that our recommendations lack sufficient contextual information. To address this, we have enhanced the discussions for each of the lessons, along with a few relevant citations (section 3.2), which we hope will in turn clarify our recommendations.
(5) Briefly discuss what Learning Management System is for recommendation 3. We briefly define what a Learning Management System is under Results, Lesson 1 (section 3.2).
(6) For title of table 1, please indicate that you used SEM for representation. We have revised the Table 1 title to indicate our use of multi-level strategies within disciplines. (section 3.1).
The revised manuscript has been seen and approved by all authors. The authors declare that they have no professional or financial conflicts of interest. The information contained in this paper has not been published elsewhere. The activities performed and data used in this paper did not involve human subjects.
We appreciate your consideration of this revised manuscript.
Reviewer 2 Report
The present case report reveals the successfully established model that it promotes awareness and education of healthcare professionals for preventing FASDs. It's informative and well-written. The content is too long for a case report, simplification should be performed before it's accepted.
Author Response
Reviewer 2
Thank you for the opportunity to revise and resubmit our manuscript titled, “Partnering with the Health Professions to Promote Prevention of an Alcohol-Exposed Pregnancy: Lessons Learned from an Academic-Organizational Collaborative”, which we hope is now suitable for publication in International Journal of Environmental Research and Public Health as a case report. We deeply appreciate the reviewers’ suggestions and have addressed them as follows:
(1) The content is too long for a case report, simplification should be performed. We have condensed the descriptions of the discipline-specific workgroup strategies and have deleted redundant information in the discussion section.
(2) Please provide the full name of “CDC” for its first appearance. Thank you for catching this error. We now provide the full name for CDC where it first appears (section 1.1).
(3) Was ethical approval needed for this study? While this manuscript does not report any data collected from human subjects, ethical approvals for the overall project were obtained through the Institutional Review Boards associated with each author’s academic institution. We now state this at the end of the Introduction (section 1).
(4) Recommendations 2 to 4 could have stronger discussions. For example, for point 4, what does “real-world” mean? Don’t we already know that public health interventions are related to real world? We agree that our recommendations lack sufficient contextual information. To address this, we have enhanced the discussions for each of the lessons, along with a few relevant citations (section 3.2), which we hope will in turn clarify our recommendations.
(5) Briefly discuss what Learning Management System is for recommendation 3. We briefly define what a Learning Management System is under Results, Lesson 1 (section 3.2).
(6) For title of table 1, please indicate that you used SEM for representation. We have revised the Table 1 title to indicate our use of multi-level strategies within disciplines. (section 3.1).
The revised manuscript has been seen and approved by all authors. The authors declare that they have no professional or financial conflicts of interest. The information contained in this paper has not been published elsewhere. The activities performed and data used in this paper did not involve human subjects.
We appreciate your consideration of this revised manuscript.
Reviewer 3 Report
The authors of this case study investigated the topic of partnering with the health professions to promote prevention of an alcohol-exposed pregnancy: lessons learned from an academic-organizational collaborative. This is an important topic (with good writing quality), and has the potential to provide guidelines for further research. I have a few minor suggestions that I hope to help the authors with minor revisions.
1. It seems to me the authors used the abbreviation “CDC” directly without providing the full name. Please provide the full name of “CDC” for its first appearance (although public health related researchers would know what it is, other non-professional readers may not have the sufficient knowledge).
2. Do authors need ethical approval for such study? If not, please mention it on the manuscript as well.
3. I think the recommendations provide good directions for further research. However, it seems to me that recommendations 2 to 4 could have stronger discussions. For example, for point 4, what does “real-world” mean? Don’t we already know that public health interventions are related to real world?
4. It could be a good idea if the authors can briefly discuss what Learning Management System is for recommendation 3.
5. For title of table 1, please indicate that you used SEM for representation.
Overall, I think this is a strong and well-written manuscript, which providers further directions for research. I would recommend for publication after the authors complete these minor revisions. Thank you.
Author Response
Reviewer 3
Thank you for the opportunity to revise and resubmit our manuscript titled, “Partnering with the Health Professions to Promote Prevention of an Alcohol-Exposed Pregnancy: Lessons Learned from an Academic-Organizational Collaborative”, which we hope is now suitable for publication in International Journal of Environmental Research and Public Health as a case report. We deeply appreciate the reviewers’ suggestions and have addressed them as follows:
(1) The content is too long for a case report, simplification should be performed. We have condensed the descriptions of the discipline-specific workgroup strategies and have deleted redundant information in the discussion section.
(2) Please provide the full name of “CDC” for its first appearance. Thank you for catching this error. We now provide the full name for CDC where it first appears (section 1.1).
(3) Was ethical approval needed for this study? While this manuscript does not report any data collected from human subjects, ethical approvals for the overall project were obtained through the Institutional Review Boards associated with each author’s academic institution. We now state this at the end of the Introduction (section 1).
(4) Recommendations 2 to 4 could have stronger discussions. For example, for point 4, what does “real-world” mean? Don’t we already know that public health interventions are related to real world? We agree that our recommendations lack sufficient contextual information. To address this, we have enhanced the discussions for each of the lessons, along with a few relevant citations (section 3.2), which we hope will in turn clarify our recommendations.
(5) Briefly discuss what Learning Management System is for recommendation 3. We briefly define what a Learning Management System is under Results, Lesson 1 (section 3.2).
(6) For title of table 1, please indicate that you used SEM for representation. We have revised the Table 1 title to indicate our use of multi-level strategies within disciplines. (section 3.1).
The revised manuscript has been seen and approved by all authors. The authors declare that they have no professional or financial conflicts of interest. The information contained in this paper has not been published elsewhere. The activities performed and data used in this paper did not involve human subjects.
We appreciate your consideration of this revised manuscript.